# Design of a Bioinspired Underwater Glider for Oceanographic Research

**DOI:** 10.3390/biomimetics8010080

**Published:** 2023-02-13

**Authors:** Diana C. Hernández-Jaramillo, Rafael E. Vásquez

**Affiliations:** 1Faculty of Science and Engineering, Southern Cross University, Coffs Harbour, NSW 2450, Australia; 2School of Engineering, Universidad Pontificia Bolivariana, Medellín 050031, Colombia

**Keywords:** marine robotics, bioinspired design, underwater glider, oceanographic research, environmental monitoring

## Abstract

The Blue Economy, which is based on the sustainable use of the ocean, is demanding better understanding of marine ecosystems, which provide assets, goods, and services. Such understanding requires the use of modern exploration technologies, including unmanned underwater vehicles, in order to acquire quality information for decision-making processes. This paper addresses the design process for an underwater glider, to be used in oceanographic research, that was inspired by leatherback sea turtles (*Dermochelys coriacea*), which are known to have a superior diving ability and enhanced hydrodynamic performance. The design process combines elements from Systems Engineering and bioinspired design approaches. The conceptual and preliminary design stages are first described, and they allowed mapping the user’s requirements into engineering characteristics, using quality function deployment to generate the functional architecture, which later facilitated the integration of the components and subsystems. Then, we emphasize the shell’s bioinspired hydrodynamic design and provide the design solution for the desired vehicle’s specifications. The bioinspired shell yielded a lift coefficient increase due to the effect of ridges and a decrease in the drag coefficient at low angles of attack. This led to a greater lift-to-drag ratio, a desirable condition for underwater gliders, since we obtained a greater lift while producing less drag than the shape without longitudinal ridges.

## 1. Introduction

Natural resources are key for the sustainable development of a growing human population [1], and the Blue Economy is being developed based on the key role of the ocean, since marine ecosystems provide assets, goods, and services [2,3], which can be capitalized in a sustainable way, as described among several targets of the Sustainable Development Goals (SDGs) [4]. This makes the ocean a new frontier for economic development [5], which poses challenges for marine ecosystems. Such challenges require a deep understanding of the ocean ecosystem due to the fact that human activities have been increasing the rate of climate change [6], inducing accelerated ocean biodiversity losses [7].

In order to fulfill the needs resulting from the emerging interest in ocean-related industries/activities, the use of modern ocean exploration techniques that use specialized tools has grown, allowing the development of climate patterns, the exploitation of energy resources, and the characterization of different marine ecosystems, among others [8,9]. Consequently, the demand for multidimensional ocean information with high temporal and spatial resolutions has also grown in order to provide quality information for decision-making processes related to marine resources [10]. Among the modern exploration technologies, one can find unmanned marine vehicles, which have been used to reach inaccessible regions all around the world [11] and which are mostly divided into three types: remotely operated (ROV) [12,13], autonomous (AUV) [14,15], and surface (USV) [16,17] vehicles.

Among underwater vehicles, AUVs navigate autonomously based on information from their surroundings, which is provided by sensors and the navigation system [14]. Such vehicles are deployed underwater and perform tasks that vary depending on the range of the area to be surveyed. Underwater gliders constitute a type of AUV that has emerged as an alternative technology to perform long-term measurements within the water column [18,19]. These vehicles move vertically through buoyancy changes and move horizontally by using wings and are useful for sustained observations needed between the coastal and open ocean [20]. Several underwater gliders are torpedo-shaped vehicles, and most recent design works have been devoted to hull optimization [21,22,23] and buoyancy change modules [24,25] in order to increase the efficiency of such long-range low-speed surveying underwater robots.

Bioinspired design is a field that has yielded successful products/processes [26] within several disciplines during the last decade, including: electric power generation [27,28], architectural design [29,30], materials science [31], medicine [32,33], and marine robotic systems [34,35,36,37,38,39], among others. Within the field of underwater vehicles, several research works have been reported during the last two decades. For instance, Font et al. [40] presented the biomimetic design and implementation of a hydrofoil propulsion system for a turtle-based AUV. Shi et al. [41] and Xing et al. [42] developed small turtle-based amphibious robots, capable of walking on land and moving underwater. Mignano et al. [43] developed a multifin biorobotic experimental platform and computational fluid dynamics simulations to understand the propulsive forces produced in a fish-like robot. Costa et al. [44] developed a series of swimming robots until they reached the carangiform locomotion style, while looking for maximum propulsive efficiency. Aparicio-García et al. [45] proposed a parallel mechanism coupled to an artificial caudal fin for a biomimetic AUV. Li et al. [46] presented the biomimetic design of an omnidirectional underwater robot with multiple tails, capable of combining different locomotion modes. Bianchi et al. [47] reported the design and experimental tests of a cownose-ray-bioinspired underwater robot. Regarding the biomimetic design of underwater gliders, Yuan et al. [48] presented the mechatronic design of a gliding robotic dolphin. Dong et al. [49] presented the biomimetic design of a whale-shark-like underwater glider, in order to combine high maneuverability and long duration. Zhang et al. [50] addressed the biomimetic design of a manta ray robot, combining gliding and flapping propulsion systems. Wang et al. [51] developed an underwater torpedo-shaped glider with a caudal fin to enable bidirectional maneuverability. Mitin et al. [52] presented a work which describes a bioinspired propulsion system based on the thunniform principle for a robotic fish, which uses a combination of elastic components with a fixed tail fin.

Gliders, among other underwater vehicles, have been of high interest for oceanographic research due to their energy efficiency and long-range sampling capabilities. Although one can find bioinspired glider designs such as the ones in [48,49,50,51], no reports have been found regarding underwater gliders inspired by leatherback sea turtles (*Dermochelys coriacea*), which are known to have a superior diving ability and to be highly adapted to pelagic swimming, thanks to the five longitudinal ridges on their carapace, which result in enhanced hydrodynamic performances [53]. Hence, this work addresses the bioinspired design of an underwater glider that can operate for several months with low energy consumption, by using the principles found in the aforementioned leatherback sea turtles. Since underwater gliders rely on battery packs for the entire mission, reducing energy consumption by increasing hydrodynamic performance could result in greater endurance and a better opportunity to collect measurements during longer periods of time. The design process covers different phases: conceptual, preliminary, and detailed design, including the selection of the instrumentation and hydrodynamic analysis. This paper is organized as follows. Section 2, Section 3 and Section 4 contain the design process, considering biomimetics and the conceptual, preliminary, and detailed design processes. Section 5 shows the underwater glider and descriptions of the mission it will perform. Finally, some conclusions and the obtained patent are presented in Section 6 and Section 7.

## 2. Underwater Glider Bioinspired Design

Biomimetics has been widely used in engineering design processes by developing solutions to problems by employing analogies with biological systems [26]. In this study, the hydrodynamic performance of an underwater glider was enhanced by the shell’s design inspired by the diving ability of leatherback sea turtles [53,54,55]. The methodology used for the biomimetic design started with the definition of the problem in terms of function, by performing a functional decomposition and transferring the engineering parameters to a biological solution. The next step was to reframe the problem in order to define a biological solution to finally extract and apply the biological principle to a technical solution [56] (Figure 1). Once the biomimetic design framework was established, the design of the underwater glider was divided into three stages, described below, covering the conceptual, preliminary, and detailed design. Such design stages were carried out using Systems Engineering (SE), an approach defined by NASA [57] as “a methodical, multidisciplinary approach for the design, realization, technical management, operations, and retirement of a system”. SE enables the interactions of components that provide functionality within a complex system that is expected to meet several requirements [58,59] and has been successfully used for the development of robotic underwater vehicles [60,61].

## 3. Conceptual Design

The first step in the underwater glider design was the conceptual design, where the user requirements were identified in the Colombian context and were translated into engineering characteristics following the Quality Function Deployment (QFD) method, as well as a functional analysis approach. This section contains the requirements’ definition and the QFD method employed to define the engineering characteristics.

### 3.1. Requirements’ Definitions

The area of concern is the coastal zone of the Colombian Caribbean and Pacific at depths up to 200 m, for seasonal scales of three months. The vehicle could be used not only to measure oceanographic variables, but also for environmental surveys, maritime surveillance, climate monitoring, to collect data to feed predictive models, and for monitoring activities in general.

Some technical requirements were taken into account in order to have a better description of the problem. The vehicle should operate with energy efficiency, and it should have the capacity to operate up to three consecutive months while measuring variables. It should fit in small boats to be transported to 60 to 120 km from the coast, and it should weigh less than 50 kg.

The vehicle will operate in a highly corrosive environment, with a pressure of 20 atm at 200 m of depth and a temperature of approximately 15 ${}^{\circ }$C. Those aspects had to be taken into account during the design process, as well as the biofouling, which may affect the vehicle during long operation periods.

### 3.2. Quality Function Deployment

The QFD method was used to organize all the information needed for a better understanding of the problem in the early design phase. The objective was to obtain measurable design targets for critical parameters that were identified from the customers’ requirements [62]. For the underwater glider design, the customers’ requirements were divided into two main categories: ground operation and water operation. The requirements were then translated into engineering specifications or engineering characteristics, which determined the target values for the design. The specifications were divided into the same categories as the customer’ requirements, and each one of them measured at least one of the customers’ needs. A relationship matrix between the customers’ requirements and the engineering specifications was developed to identify the importance of each specification and to determine the design targets, which in order of relevance were defined as: low operational cost, low energy consumption, long battery life, long traveled distance, and long operating time.

### 3.3. Preliminary Design Specifications

A Preliminary Design Specifications document (PDS) was developed as a result of the QFD analysis. From this analysis, it was expected to have a bioinspired design of the vehicle to find a nature-related solution to the efficiency in the motion pattern during the operation time, while performing dives to 200 m of depth at an approximate velocity of 0.2 m/s. The operating environment for the glider is saltwater, which is highly corrosive. Besides, at 200 m of depth, the pressure is approximately 20 atm at 15 ${}^{\circ }$C. As was defined before, the vehicle’s weight should be less than 50 kg, and the length should be less than 2 m. The energy sources were battery packs such as those used for the vehicles described in [20]. The vehicle should be easily transported in small boats and should be suitable for operations in different sea conditions. The calibration of the sensors was an important aspect to take into account for mission planning, as well as the maintenance requirements of the components to ensure reliability. The target market in Colombia is centered on oceanography and marine biology research centers, private and governmental companies, and military forces. Possible applications include environmental surveys, maritime surveillance and reconnaissance, climate monitoring, and environmental impact.

### 3.4. Functional Architecture

In order to accomplish the main purpose of performing measurements of the water column for long periods of time and at large spatial scales, six basic functions were defined for the underwater glider (Figure 2). These functions can be described as follows:To measure oceanographic variables: This function represents the principal purpose of the underwater glider to be designed. It is in charge of performing the measurements of oceanographic variables such as salinity, temperature, depth, and dissolved oxygen.To move the device: This function is in charge of moving the vehicle in the water, following a sawtooth pattern reaching up to a 200 m depth.To supply energy: This function is responsible for providing energy to all components to guarantee proper operation during the mission.To protect the system: This function represents the task of protecting the vehicle and all of its components from the environment and against failures.To perform the mission: This function is in charge of accomplishing the mission including preparing, transporting, launching, and recovering the vehicle.To manage information: This function is responsible for managing all the information needed for navigation and all the information collected during operation.

The top-level functions were divided into subfunctions (Figure 3) that describe, at a lower level, what actions are needed to achieve the system’s objectives. Additionally, interactions between functions and external factors that may affect the performance of the vehicle were also defined.

## 4. Preliminary Design

The purpose of the preliminary design was to develop a general layout of devices that perform the functions and meet the prescribed requirements. Every function may be solved with several alternatives for the software and hardware. These solutions were the result of brainstorming sessions, and they were analyzed in terms of feasibility, technological maturity, and relation to the design targets. The best candidates were selected in order to achieve a design solution for the problem. During the preliminary design, the systems were described in terms of components, and the interfaces between them were defined to obtain a physical architecture. The selection and design process of the solutions for the underwater glider are presented in this section.

### 4.1. Oceanographic Measurements

The function of measuring oceanographic variables includes the measurement of salinity, temperature, depth, and dissolved oxygen. Commercial solutions were proposed to accomplish this function. At least three alternatives were analyzed, and a selection matrix was used to choose the best option to fulfill the design requirements. The selected options integrated in the underwater glider were the Glider Payload (CTD-GPCTD) from Sea-Bird Electronics designed for a 350 m depth, along with the SBE 43F Dissolved Oxygen (DO) sensor from the same company to facilitate the integration of the two sensors.

### 4.2. Vehicle Motion

The motion of the underwater glider consisted of a vertical component due to changes in buoyancy and a horizontal component due to the lift generated by the wings. This section describes the positioning and buoyancy systems and the wing sizing, and then, it is centered on the bioinspired shape design for the vehicle, as well as the hydrodynamic analysis of the configuration.

#### 4.2.1. Positioning System

The underwater glider needs to be positioned every time it goes to the surface in order to adjust the trajectory. To solve this function, a GPS 15xL from Garmin was selected. This global positioning system permits locating the vehicle by receiving satellites signals and decoding them to calculate the position in appropriate coordinates. The GPS receiver requires a source of power, an active GPS antenna, and a clear sight to satellite signals while at the surface. That is why the vehicle needs to adjust its attitude to maintain the antenna out of the water while positioning. The UM7 from CH Robotics was selected as the Attitude and Heading Reference System (AHRS). This device employs a triaxial accelerometer, a rate gyro, and a magnetometer with an extended Kalman filter to estimate the attitude and heading. The OS5000-S 3 axial digital compass from Ocean Server was selected to provide heading, roll, and pitch data to assist navigation. Finally, the PA200 digital precision altimeter from Tritech was chosen to measure the seafloor distance.

#### 4.2.2. Buoyancy System

The subfunctions of going down and coming up to the surface can be solved simultaneously by performing changes in the buoyancy force using ballast tanks, buoyancy engines with water, oil or air pumps and bladders, or by using thermal engines such as the one used in the Slocum Glider [19,63,64]. In this case, the hydraulic power unit was selected to change the buoyancy by moving oil from an internal reservoir to an external flexible bladder. The reversible hydraulic power pack (HPR 1105HPRNSS02) from Hydra Products was chosen. This consists of a self-contained DC motor, gear pump, reservoir, internal valving, load hold checks, and relief valves. This hydraulic pack has a tank of 0.2 L, which corresponds to the volume needed for the external bladder.

#### 4.2.3. Wing Sizing

Underwater gliders use wings to generate lift with a horizontal component due to the attitude of the vehicle while gliding. These wings operate at a low Reynolds number due to the slow operation velocity. The Reynolds number helps to predict the flow pattern depending on the ratio between inertial and viscous forces acting in a body immersed in a fluid. The appropriate selection of a hydrofoil is important because it generates the necessary forces to move the vehicle through the water. Symmetrical airfoils such as the NACA 00 series are used as hydrofoils for underwater gliders, since they can generate lift with the same lift-to-drag ratio (L/D) in both descending and ascending states [65]. Symmetrical airfoils have the disadvantage of having a lower L/D ratio compared with cambered airfoils. A higher L/D is ideal for an efficient glide, but in order to use a cambered airfoil, the glider should be able to invert the flight at the different states of the glide pattern or it should use high-lift devices such as flaps and slats, which increase the complexity of the design.

For the selection of the hydrofoil, it was desired to have a maximum lift coefficient and a proper ideal lift coefficient, which corresponds to the coefficient at the angle of attack at which the drag coefficient does not have representative variations. Besides, the lowest minimum drag was also desired, as well as a high ratio between the lift and drag coefficients. The design lift coefficient corresponded to the highest $C_{L}/C_{D}$. A hydrofoil with a higher slope in the function of the lift coefficient vs. the angle of attack has a better performance, and it is preferred to have a gentle drop in the lift after a stall for a safer performance. For the underwater glider, an Eppler E171 airfoil was selected; this airfoil has a symmetrical section, which fulfills the requirements at different low Reynolds numbers. The maximum thickness is 12.3% at 32.4% of the chord. The wing design target was to maximize the lift (*L*) while minimizing the drag (*D*) and pitch moment (*M*). Based on [66], the design began with a platform area of S=0.15 m${}^{2}$ as a first approximation. Lifting line theory was used to calculate the lift distribution along the span and the total lift wing coefficient [67]. The variation of the segment’s lift coefficient along the semispan is shown in Figure 4.

The taper ratio, λ, is the ratio between the tip chord $C_{tip}$ and the root chord $C_{root}$ and takes values from 0 to 1. The use of a taper ratio reduces the induced drag and improves the wing lift distribution. Additionally, it reduces the wing weight, produces a lower bending moment at the wing root, and improves lateral control. Based on [66,68], a taper ratio of 0.7 was assumed, and the chord at the root and the tip of the wing was calculated as follows:(1)$$C_{root}=\frac{2S}{b(1+\lambda )},$$
(2)$$C_{tip}=\lambda C_{root},$$
which yields a root chord of 0.1765 m and a tip chord of 0.1235 m.

#### 4.2.4. Tail Sizing

The primary purpose of the vertical tail is to counteract the moment produced by sideslip forces acting on the vehicle and wings. The tail was sized with respect to the baseline fixed wing following an empirical method proposed by Raymer [69]. This method uses a tail volume coefficient $C_{vt}$, which depends on the aircraft wing planform area *S*, the wing span *b*, the distance between the aerodynamic center of the wing and the aerodynamic center of the vertical tail $L_{vt}$, and the vertical tail planform area $S_{vt}$. $C_{vt}$ can be computed as
(3)$$C_{vt}=\frac{L_{vt}S_{vt}}{bS}$$

A typical value of $C_{vt}$ for sailplanes is 0.02 [69], which is relatively small compared to the tail volume coefficient for current underwater gliders [70]. The tail volume coefficient was estimated as 0.25 for the underwater glider, and it was used to calculate the tail area with Equation (Equation 3). The root and tip chords for the vertical tail were calculated with Equations (Equation 1) and (Equation 2), with a tail span of 0.5 m and a taper ratio of 0.7.

#### 4.2.5. Biomimetic Analysis

Biomimetics has been widely used in engineering design processes by developing solutions to problems employing analogies with biological systems [26]. Methods such as the direct approach, the case study method, BioTRIZ based on Altshuller’s theory of inventive problem solving (TRIZ), the functional modeling, and biological analogy search tools [71] have been used to inspire design concepts and solve engineering problems. In this case, biomimetics was used to obtain a bioinspired design of the external shape of the underwater glider, in order to enhance the hydrodynamic performance, based on the steps proposed by Helms et al. [56], which are described in Figure 1. The leatherback turtle (*Dermochelys coriacea*) was selected as the natural referent to be used in the underwater glider design, due to the superior diving ability with energy efficiency, the long-distance migration, and the V-shaped pattern of diving. The longitudinal ridges on their shells can reduce drag and increase lift due to the delayed flow separation [53]. This feature can be adapted to the design of an underwater vehicle with a better hydrodynamic performance. To design the shell, images from 18 turtles were analyzed, extracting from them the basic curves representing their shapes. The curves were described as Cartesian coordinates and normalized based on the overall length. Then, 11 of those turtles were selected based on the information that could be extracted from them. An interpolation from the curves was performed to obtain a shape as similar as possible to the typical body of leatherback turtles. Both the curves on the longitudinal axis and the ridges were parametrized to obtain a smooth profile. The design of the external shape of the glider consisted of the parametrization of the basic shape of a turtle shell (Figure 5).

The shape in Figure 5a represents the parametrization for the *x*–*y* plane, which is given by
(4)$$x_{t}=2t_{x}\left[0.5065\sqrt{\frac{y}{l}}+0.2469\left(\frac{y}{l}\right)-1.5709{\left(\frac{y}{l}\right)}^{2}+1.7052{\left(\frac{y}{l}\right)}^{3}-0.8898{\left(\frac{y}{l}\right)}^{4}\right],$$
where $x_{t}$ and *y* are the *x*–*y* coordinates, respectively, $t_{x}$ is the maximum thickness in the *x* direction, and *l* is the total length of the vehicle; in this case, *l* = 1.5 m; the *x* axis is aligned with the sway motion of the vehicle; the *y* axis is aligned with the surge; the *z* axis corresponds to the heave motion. The shape in the *y*–*z* plane (Figure 5b) is described by
(5)$$z_{t}=2t_{z}\left[0.5065\sqrt{\frac{y}{l}}+0.2469\left(\frac{y}{l}\right)-1.5709{\left(\frac{y}{l}\right)}^{2}+1.7052{\left(\frac{y}{l}\right)}^{3}-0.8898{\left(\frac{y}{l}\right)}^{4}\right],$$
where $z_{t}$ is the *z* coordinate and $t_{z}$ represents the maximum thickness in the *z* direction.

Additionally, the design included longitudinal ridges in both the upper and lower sections of the vehicle, inspired by the ridges of leatherback turtles. The cross-section on the body in a particular *y* coordinate corresponds to an ellipse given by the thickness $x_{t}$ as the principal axis and the thickness $z_{t}$ as the secondary axis:(6)$$x=x_{t}\sin\left(\theta_{e}\right),$$
(7)$$z=z_{t}\cos\left(\theta_{e}\right),$$
where $\theta_{e}$ is the angle used to determine a quarter of the ellipse; it goes from 0 to π/2.

The number of ridges in the quarter of the ellipse is given by *N*, while $\theta_{N}$ defines the angle between each ridge and depends on *N*. The coordinates of the ridges at a given angle were determined by (Equation 6) and (Equation 7). The quadratic Bézier curve through the three points presented was used to describe the curve between two consecutive peaks as follows:(8)$$B\left(t\right)={\left(1-t\right)}^{2}P_{0}+2t\left(1-t\right)P_{1}+t^{2}P_{2},$$
where *t* goes from 0 to 1 and $P_{i}$, with *i* from 0 to 2, corresponds to the control points given by two consecutive peaks and an auxiliary point determined by a factor *r* of the maximum thickness of every cross-section. The coordinates of the auxiliary point are given by
(9)$$x_{s}=rx_{t}\sin\left(\theta_{s}\right),$$
(10)$$z_{s}=rz_{t}\cos\left(\theta_{s}\right),$$
where $\theta_{s}$ is the angle between two auxiliary points and corresponds to the midpoint between two consecutive peaks. The cross-section of the parametric shape of the vehicle is presented in Figure 6.

The parametrization used to described the profiles of the glider allowed performing changes on the overall shape of the shell according to the basic dimensions of the vehicle. This gave the opportunity for further hydrodynamic analyses and a comparison of the longitudinal ridges’ performance in different conditions.

#### 4.2.6. Hydrodynamic Analysis

Computational Fluid Dynamics (CFD) was used to predict the hydrodynamic characteristics of the underwater glider. These characteristics included the lift and drag forces, the moments, and the corresponding coefficients. ANSYS Fluent® was used to solve the Reynolds-Averaged Navier–Stokes (RANS) equations for the conservation of mass and momentum in a steady state and immersed in an incompressible flow (Figure 7 and Figure 8). The Reynolds number was lower than other underwater vehicles that operate at higher speeds, and considering that the fluid was seawater, the flow regime was turbulent. Therefore, the standard κ−ϵ turbulence model was used along with an enhanced wall treatment to model the near-wall region. We used a pressure-based solver and a SIMPLE scheme for pressure–velocity coupling. For this model, the non-dimensional wall parameter $y^{+}$ should be within the constraints of the viscous sublayer ($y^{+}<1$).

Simulations were carried out for the parametric shape without ridges and for the shape with ridges at several angles of attack (2, 4, 6, 10, 14, 18, and 22 degrees) and at velocities of 0.1, 0.2, and 0.3 m/s. Figure 9a shows an increase in the lift coefficient due to the effect of ridges in the flow distribution for every velocity analyzed. A decrease in the drag coefficient at low angles of attack (Figure 9b) was observed due to the effect of the longitudinal ridges, despite the fact that, at greater angles of attack, there was an increase in the drag coefficient. The increase in the lift coefficient resulted in a greater lift-to-drag ratio (L/D), which is a desirable condition for the design of underwater gliders, making it possible to obtain a greater lift at small angles of attack while producing less drag than the shape without ridges (Figure 10). The results agreed with the analysis presented by Bang et al. [53] for positive angles of attack.

A greater lift-to-drag ratio can be translated into a higher hydrodynamic efficiency at smaller angles of attack. Glider efficiency is related, among other things, to the mechanical work applied to obtain optimal changes in the attitude of the vehicle; therefore, it is directly linked to energy consumption. Since underwater gliders rely on battery packs for the entire mission, reducing energy consumption by increasing hydrodynamic performance could result in greater endurance and a better opportunity to collect measurements during longer periods of time.

### 4.3. Attitude System

Sliding masses are used in gliders to control the pitch angle by changing the position of the center of gravity relative to the center of buoyancy during dives and climbs. It is common to use battery packs as the sliding and rotating masses to induce roll and course change. The PA-08 mini track actuator was selected to generate the linear motion of the battery packs and the consequent pitch angle change, while a gear motor of 168 rpm was selected to generate the roll. The mini track actuator has a stroke of 50.8 mm (2 in), a force of 50 lb, and a speed of 1.18 in/s. The track designs implies a small size since the stroke does not extend or retract from the unit.

### 4.4. Energy Supply

To supply energy to all components of the device, the maximum power consumption (W), input voltage (V), and current drain (A) of every component were analyzed. A rechargeable battery pack of Lithium Nickel Manganese Cobalt Oxide (LiNiMnCo) of 14.8 V was selected to satisfy the components’ needs, with 8 Ah and 118.4 Wh. A LiNiMnCo battery pack was chosen because it provides a higher energy density (mAh/weight) with a lower cost and a long cycle life. However, this type of battery needs to be properly used to guarantee safety. The next step was to calculate the battery pack’s capacity based on how long the vehicle will operate, and based on this information, 21 battery packs were used to satisfy the power needs. This is an important consideration, since a long operation time was desired by the customers. The batteries have 8.95 MJ of energy, which meets the energy available in commercial underwater gliders.

### 4.5. Information Management

During operation, the underwater glider will have to manage data collected with oceanographic sensors, as well as the satellite signals for positioning. Additionally, it will receive information of internal variables to monitor the operating conditions. All the information will be received, processed, stored, and transmitted. The main component selected to solve this function was the Fox embedded computer designed by VersaLogic powered by a DMP Vortex86DX2 processor. The main requirements to select the computer were a low power consumption, a small size, and versatility to allow the integration of current and future sensors including serial communication channels, as well as analog and digital channels. While underwater, the information will be stored in a removable micro SD card solid-state drive supported by the embedded computer and in an external hard drive. At the surface, the glider will need to transmit information to the operation center. To solve this task, the Iridium system was chosen as the satellite-based telemetry system. The Iridium Short Burst Data (9603 SBD) transceiver provides the capability of monitoring and exchanging data with remote devices deployed in places beyond terrestrial wireless connections.

### 4.6. Protection System

The system will need protection against failures and protection from the environment and will need to act in case of failures or emergencies. The main solution to solve this function was the structural hull, which consisted of a pressure vessel used to protect the electronic components from seawater and water pressure. The structural design included the hull design, material selection, and sealing methods. Another component to protect the system was the use of sensors to measure internal variables such as the temperature and humidity. An electronic board designed by [60] was used to measure the internal temperature, flooding, internal humidity, and energy consumption. This board was connected to the selected embedded computer to monitor the variables and perform actions in case of an emergency.

#### 4.6.1. Hull Design

The pressure vessel to protect the electronic components was designed to resist a water pressure of 2 MPa corresponding to 200 m of operational depth. The structure was divided into 3 hulls, one at the front as the electronic bay, another one in the middle to house the battery packs, and the last one in the rear position for the buoyancy engine. The electronics bay had a diameter of 0.14 m and a length of 0.215 m. Because of the battery packs’ dimensions and the space available from the parametric design of the external hydrodynamic shape, the battery bay had an external diameter of 0.225 m and a length of 0.665 m. Finally, the buoyancy control bay had a diameter of 0.14 m and 0.3 m of length. The purpose of the hull design was to select an appropriate material and to find the appropriate geometry and wall thickness that satisfied the design requirements.

The method proposed by Ashby [72] was used for the material selection of the hulls, following a pressure vessel case study. The first step was to translate the design requirements into functions, constraints, objectives, and free variables. The main function of the vessel was to support a determined pressure. The established constrains were the external diameters and lengths. The objective was to maximize safety using the yield strength or to maximize safety using leaking before the break condition. The free variable for this case was the choice of the material. In this selection, stainless steels, low alloy steels, cooper, aluminum alloys, and titanium alloys were found suitable to fabricate the desired pressure vessel since they are commonly used for the hull fabrication of underwater vehicles [73]. Among such materials, stainless steel exhibits high strength, but a heavy weight, and it is necessary to treat the surface in order to use it in sea water. Aluminum alloys exhibit a light weight and high strength and do not become magnetized, but a surface treatment is also needed to use them in underwater operations. Finally, titanium alloys exhibit a light weight, a high strength, and a high corrosion resistance, but they are expensive [74]. Aluminum alloys Series 5000, 6000, and 7000 are used for hull structures in underwater vehicles, especially the 6000 series, the performance in strength and corrosion of which is in the between the 5000 and 7000 series. The typical surface treatments for aluminum alloys include anodizing, electrolysis nickel plating, electroless nickel plating, and painting.

For the three hulls of the vehicle, the AA6061-T6 aluminum alloy was selected. This material has a yield strength of 276 MPa. The wall thickness by yield criteria ($t_{y}$) for every hull was approximated with a safety factor, $S_{f}$, of 2, and the hoop stress $\sigma_{y}$, for the given radius *r* at the working pressure *p*, as follows:(11)$$t_{y}=\frac{2pr}{\sigma_{y}}.$$

Additionally, the buckling criteria were considered to obtain the wall thickness ($t_{bc}$) using
(12)$$t_{bc}\geq d{\left[\frac{S_{f}pl}{2.59ED}\right]}^{0.4},$$
where *E* is the elastic modulus of the material, 68,900 MPa for AA6061-T6, *d* is the hull diameter, and *l* is the length of the hull. Table 1 shows the wall thickness by the yield and buckling criteria for each hull. The size of the wall thickness was determined by the circumferential buckling criteria because it is greater than the wall thickness determined by the yield criteria, which means that buckling is the predominant failure mode. The wall thicknesses of Hull 1 and Hull 3 were set to be equal in order to ease the manufacturing, because they had the same external diameter.

#### 4.6.2. Sealing Analysis

Sealing is an important issue for underwater vehicles, due to the use of electrical and electronic components, which need to be isolated from the water to function properly. O-rings are the most-common sealing methods. They consist of circular cross-section rings molded from elastomeric or thermoplastic materials. The O-ring is contained in a groove, where it is deformed when pressure is applied. The end caps of the hulls were determined to be piston seal caps that fit inside the hulls. For this type of cap, water pressure compresses the diameter of the hull slightly, squeezing the O-ring and generating a waterproof seal. The Parker O-Ring Handbook [75] was used to select the O-ring used in the hull design and to dimension the grooves that were to be machined in caps. Nitrile-Butadiene Rubber (NBR) was selected as the O-ring’s material due to its commercial availability and its capacity to work with seawater. The internal diameter of the wall was the parameter used to size and select the O-ring. It was decided to use two O-rings to ensure a better sealing.

#### 4.6.3. End Caps’ Design

The flat end caps for Hull 1 and Hull 3 must be able to resist deflection due to the pressure of the water at the operational depth. The bending stress was calculated assuming that there was constant pressure along the flat circular plate of constant thickness using
(13)$$\sigma =\frac{6M}{t^{2}},$$
where *M* is the bending moment and *t* is the end cap thickness.

The moment at the center $M_{c}$ and the moment at the reaction $M_{ra}$ were calculated with the following equations, for uniformly distributed pressure with fixed supports:(14)$$M_{c}=\frac{pr^{2}\left(1+\nu \right)}{16},$$
(15)$$M_{ra}=\frac{-pr^{2}}{8},$$
where ν is the Poisson ratio of the material, which in this case corresponds to the AA6061-T6 aluminum alloy. The minimum thickness of the end cap to withstand the working pressure was calculated, with a safety factor of 2, using
(16)$$t_{min}=\max\left(\sqrt{\frac{6S_{f}M_{c}}{\sigma_{y}}},\sqrt{\frac{6S_{f}M_{ra}}{\sigma_{y}}}\right).$$

The minimum thickness was 7 mm for the end caps to be used in Hull 1 and Hull 3. For the caps to be used to join Hull 2 with the other hulls, the bending moment at the reaction was calculated as follows:(17)$$M_{ra}=\frac{-p}{8R^{2}}{\left(r^{2}-r_{0}^{2}\right)}^{2},$$
where $r_{0}$ is the inner ratio of the cap. The thickness for this caps was 9.7 mm.

## 5. Bioinspired Glider

The underwater glider consisted of a bioinspired external shell with longitudinal ridges used to enhance hydrodynamic performance. Wings were used to generate forward motion due to the lift force, while a vertical tail was used to counteract the moments produced by sideslip forces acting on the vehicle (Figure 11a). Internally, three hulls were used to house the components (Figure 11b). The first hull corresponded to the electronics bay. It housed the embedded computer and the navigation, positioning and communication instruments such as the AHRS, the compass, the GPS receiver, the Iridium modem, and the electronic board to measure the internal variables. The second hull was the battery bay, which housed the battery packs and pitch and roll mechanisms. The final hull was the buoyancy bay, which housed the buoyancy engine. Figure 12 shows a scheme of the internal distribution of the electronics, batteries, and buoyancy bays. Finally, CTD, DO sensors, and the altimeter needed to be located in a flooded bay to measure the water conditions and distance to the seabed (Table 2). The physical architecture of the underwater glider is presented in Figure 13.

### 5.1. Mathematical Model

A mathematical model that represents the vehicle’s dynamics was derived based on the works from Leonard and Graver [76] and Graver [77], who used such a model for the control of the ROGUE underwater vehicle. Initially, the glider was assumed as a rigid body immersed in a fluid. The vehicle was controlled by changes in buoyancy and the movement of internal masses [77].

Figure 14 shows a fixed coordinate system that was located at the buoyancy center of the vehicle. Axis 1 was aligned with the surge motion of the vehicle, positive pointing to the nose; Axis 2 was aligned with the wings’ axis, which corresponds to the sway motion; Axis 3 was chosen to be orthogonal to the plane of the wings, positive pointing downwards, and corresponds to the heave motion.

In order to obtain the equations of motion, we first describe the relationship between the total mass and internal masses, which control the buoyancy, as follows:(18)$$m_{v}=m_{h}+m_{w}+m_{b}+\bar{m}=m_{s}+\bar{m},$$
where the total mass of the vehicle $m_{v}$ is given by the sum of a stationary mass $m_{s}$ and the moving internal mass $\bar{m}$. The stationary mass is the relationship between the hull mass $m_{h}$, which is distributed uniformly throughout the vehicle, a fixed point mass $m_{w}$, which may be offset from the buoyancy center, and the ballast mass $m_{b}$, which is variable. Finally, the net buoyancy $m_{0}$ is given by $m_{v}-m$, where *m* is the mass of the displaced fluid [77]. For this analysis, the motion of the vehicle was restricted to the vertical plane and was determined by the change of the vehicle’s position in the *x* axis, $\dot{x}$, and the change of the vehicle’s position in the *z* axis, $\dot{z}$, given by
(19)$$\dot{x}=v_{1}\cos\left(\theta \right)+v_{3}\sin\left(\theta \right),$$
(20)$$\dot{z}=-v_{1}\sin\left(\theta \right)+v_{3}\cos\left(\theta \right),$$
where θ is the pitch angle, and its rate of change $\dot{\theta }$ is given by
(21)$$\dot{\theta }=\Omega_{2}.$$

The rate of change of the angular velocity computed in the body coordinate system, $\dot{\Omega }$, and the change of Components 1 and 3 of the velocity computed in the body coordinate system, ${\dot{v}}_{1}$ and ${\dot{v}}_{3}$, are described by
(22)$${\dot{\Omega }}_{2}=\frac{1}{J_{2}}\left(\left(m_{3}-m_{1}\right)v_{1}v_{3}-\bar{m}g\left(r_{p1}\cos\left(\theta \right)+r_{p3}\sin\left(\theta \right)\right)+M_{DL}-r_{p3}u_{1}+r_{p3}u_{3}\right),$$
(23)$${\dot{v}}_{1}=\frac{1}{m_{1}}\left(-m_{3}v_{3}\Omega_{2}-P_{p3}\Omega_{2}-m_{0}g\sin\left(\theta \right)+L\sin\left(\alpha \right)-D\cos\alpha -u_{1}\right),$$
(24)$${\dot{v}}_{3}=\frac{1}{m_{3}}\left(-m_{1}v_{1}\Omega_{2}-P_{p1}\Omega_{2}-m_{0}g\sin\left(\theta \right)-L\sin\left(\alpha \right)-D\cos\alpha -u_{3}\right),$$
where $J_{2}$ represents the second diagonal element of the inertia matrix of the vehicle and $m_{1}$ and $m_{3}$ are the first and third diagonal elements of the sum of the body and added mass. $u_{1}$ and $u_{3}$ are the first and third components of the control vector, and $u_{4}$ is the controlled variable mass rate. $P_{p1}$ and $P_{p3}$ are the first and third components of the linear momentum computed in the body coordinate system, and $r_{p1}$ and $r_{p3}$ are the corresponding components of the moving mass position computed in the coordinate system. α is the angle of attack given by $\cos\left(\alpha \right)=v_{1}/\sqrt{v_{1}^{2}+v_{3}^{2}}$. *L* is the lift force. *D* is the drag force. $M_{DL}$ is the viscous moment.

The changes of Components 1 and 3 of the moving mass’s position computed in the body coordinate system, ${\dot{r}}_{p1}$ and ${\dot{r}}_{p3}$, are given by
(25)$${\dot{r}}_{p1}=\frac{1}{\bar{m}}P_{p1}-v_{1}-r_{p3}\Omega_{2},$$
(26)$${\dot{r}}_{p3}=\frac{1}{\bar{m}}P_{p3}-v_{3}-r_{p1}\Omega_{2}.$$

The rate of change of the linear momentum components, ${\dot{P}}_{p1}$ and ${\dot{P}}_{p3}$, is given by
(27)$${\dot{P}}_{p1}=u_{1},$$
(28)$${\dot{P}}_{p3}=u_{3},$$
while the change of the ballast’s mass ${\dot{m}}_{b}$ is given by
(29)$$\dot{m_{b}}=u_{4}.$$

The glide path angle ξ is given by ξ=θ−α. The velocity of the vehicle is determined by $V=\sqrt{\left(v_{1}^{2}+v_{3}^{2}\right)}$. A trajectory can be specified by obtaining the desired glide path angle $\xi_{d}$ and the desired velocity $V_{d}$. Both are defined in an inertial coordinate $\left(x^{\prime },z^{\prime }\right)$, where $x^{\prime }$ coincides with the position along the desired path and $z^{\prime }$ gives the vehicle’s position with respect to the perpendicular distance to the desired path. Gliding control in the vertical plane consists of the control direction and the speed of the vehicle’s glide path, as well as to the control of the gliding along a prescribed line where the dynamics of $z^{\prime }$ is given by
(30)$${\dot{z}}^{\prime }=\sin\left(\xi \right)\left(v_{1}\cos\left(\theta \right)+v_{3}\sin\left(\theta \right)\right)+\cos\left(\xi \right)\left(-v_{1}\sin\left(\theta \right)+v_{3}\cos\left(\theta \right)\right).$$

### 5.2. Mission Description

A standard mission is described in Figure 15. It starts with pre-launch tasks, which include mission planning based on the environment conditions, endurance, and science requirements (path and sampling). It also includes transporting the glider to the field in a shipping case, assembling the vehicle, and performing calibration, ballasting, battery and communication checks, and other tests to make sure all systems are working as expected. The launch procedure will be carried out using a cradle cart to slip the glider into the water from a small boat or using a crane or winch to launch/recover it from larger boats.

At the surface, the vehicle should change its pitch angle in order to raise the tail and expose the antenna to allow GPS positioning and then Iridium communication with the operation center, then the GPS positioning will be obtained again to fix the position before every dive. A dive is performed until the operational depth is achieved or until an abort condition is reached. In this phase, adjustments of the buoyancy and pitch angle are made to be negatively buoyant and to obtain the attitude needed to reach the prescribed depth at the desired operational velocity. When the working depth is reached, the glider will start a smooth transition from the dive to the climb condition, avoiding a stall. The transition includes changes from the negative to the positive buoyant condition and moving masses to obtain the desirable pitch angle to go up to the surface. Data sampling will be performed throughout the dive and climb phases. When the vehicle reaches the surface, it starts its positioning and communication phases until it enters in a new diving cycle or until it enters in a recovery phase. A command from the operation center, the completion of the mission, or a detected error condition will cause the recovery of the vehicle from a boat. A maintenance procedure will be required after every mission, including cleaning the vehicle with fresh water and drying it with a soft cloth. It will be required to calibrate the compass every time the battery packs are replaced, and other sensors will require calibration before and after each deployment.

## 6. Conclusions

This paper addressed the design process of an underwater glider to be used in oceanographic research. The design was inspired by leatherback sea turtles (*Dermochelys coriacea*), which have great diving ability and exhibit enhanced hydrodynamic performance. The design process combined elements from Systems Engineering and bioinspired design approaches. The process was divided into three phases, and the conceptual design was used to establish preliminary specifications of the vehicle, which yielded the functional architecture that describes the desired performance. Then, the preliminary and detailed design stages considered different alternatives to fulfill the prescribed requirements and provided a general layout of the vehicle with the help of the biomimetic design using analogies with the selected biological system and the functional decomposition.

The longitudinal ridges of the leatherback turtle were used in the design of the underwater vehicle, and computational fluid dynamics was used to simulate its hydrodynamic performance. It was found that the ridges increased the lift coefficient and decreased the drag coefficient at low angles of attack, which led to a greater lift-to-drag ratio. For an underwater glider, this is a desirable condition that makes it possible to obtain a greater lift while producing less drag than a shape without longitudinal ridges.

The detailed design of the vehicle was developed by considering system integration, taking into account the interactions among the components and the interaction with the environment, as described by the functional and physical architecture diagrams. Although we obtained the physical architecture for and details of the glider’s construction, the work was limited to obtaining a virtual prototype, and further work is needed to plan the fabrication and molding requirements to manufacture the shell, the assembly, and the testing of the vehicle in real conditions.

## 7. Patents

The underwater glider’s design process resulted in a Colombian patent for the bioinspired shell: NC2019/0008406 [78] (Figure 16).

## Figures and Tables

**Figure 1 biomimetics-08-00080-f001:**
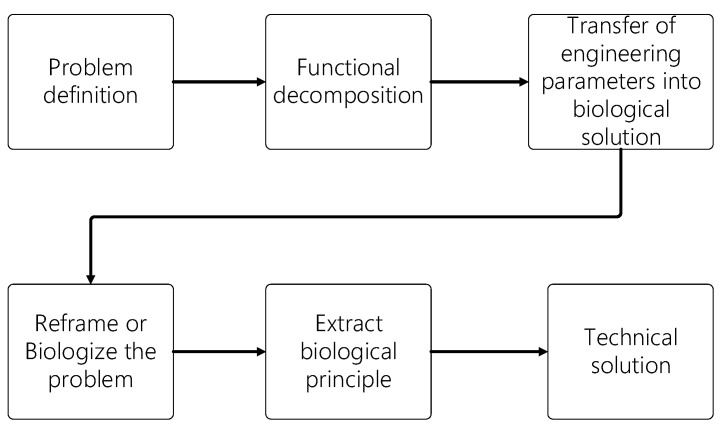
Biomimetic design process. Diagram created with the steps described in [56].

**Figure 2 biomimetics-08-00080-f002:**
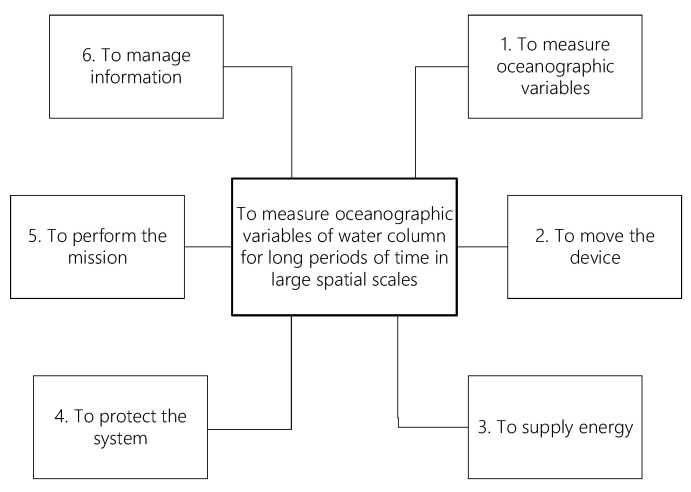
Main functions of the system.

**Figure 3 biomimetics-08-00080-f003:**
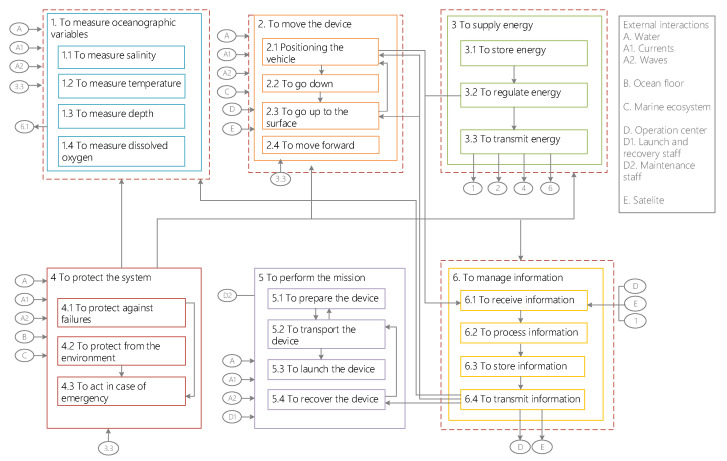
Functional architecture. Main functions are divided into subfunctions. Arrows that enter into a function indicate that other functions affect this function. Arrows that go out of a function indicate that this function affects other functions.

**Figure 4 biomimetics-08-00080-f004:**
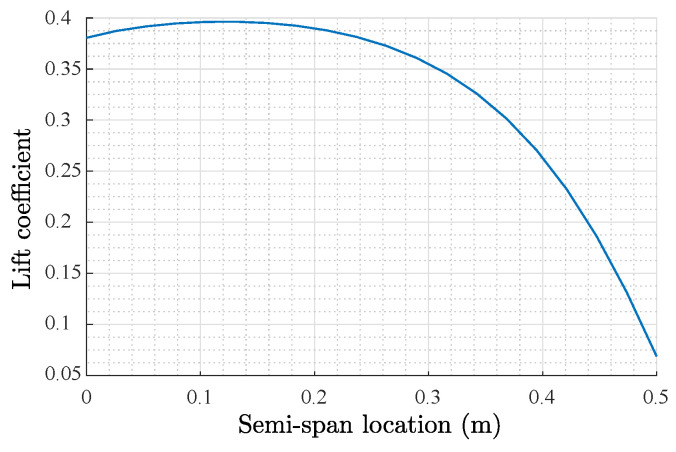
Lift distribution along the semispan.

**Figure 5 biomimetics-08-00080-f005:**
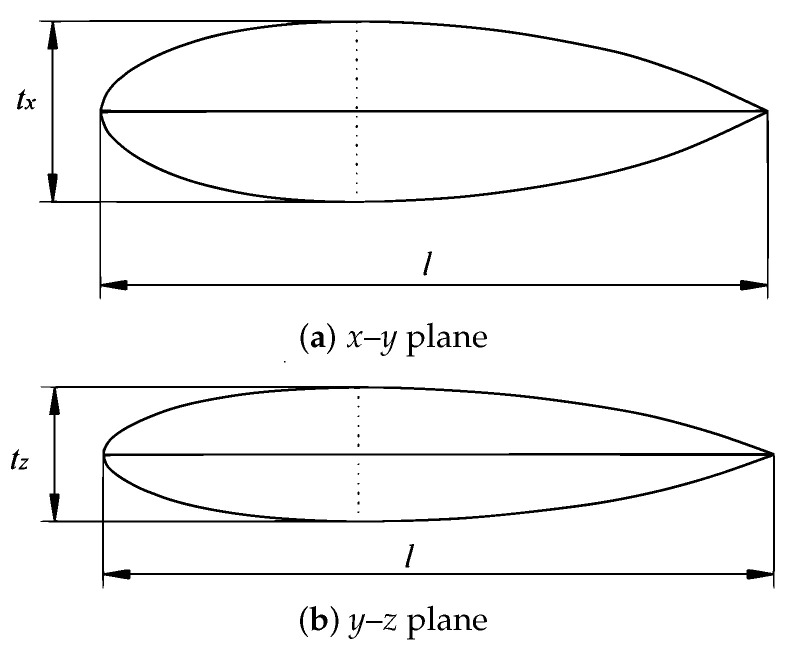
Parametric curves in two planes. Proposed by the authors based on the leatherback turtle’s geometry [53].

**Figure 6 biomimetics-08-00080-f006:**
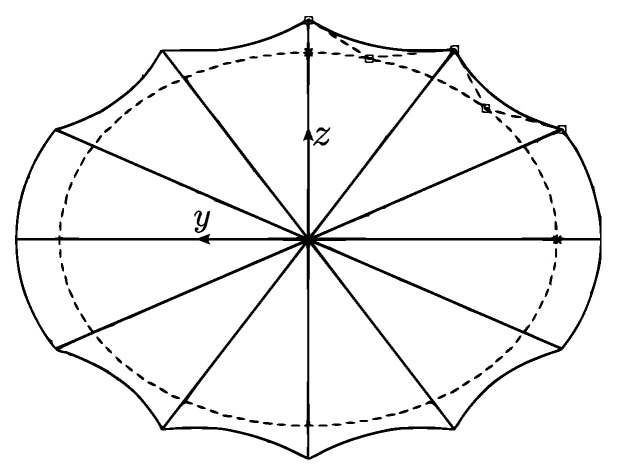
Cross-section of the parametric shape.

**Figure 7 biomimetics-08-00080-f007:**
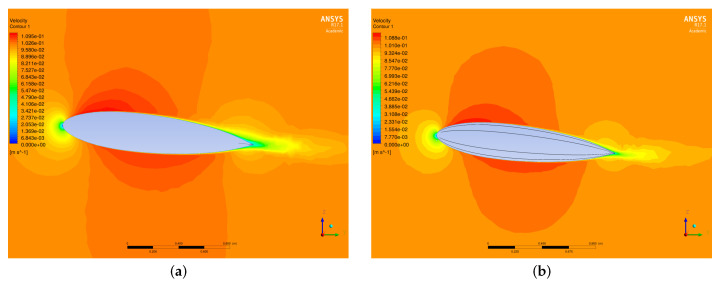
Velocity contour. (**a**) Parametric shape without ridges. (**b**) Parametric shape with ridges. Simulations performed using ANSYS Fluent^®^ R17.1 [55].

**Figure 8 biomimetics-08-00080-f008:**
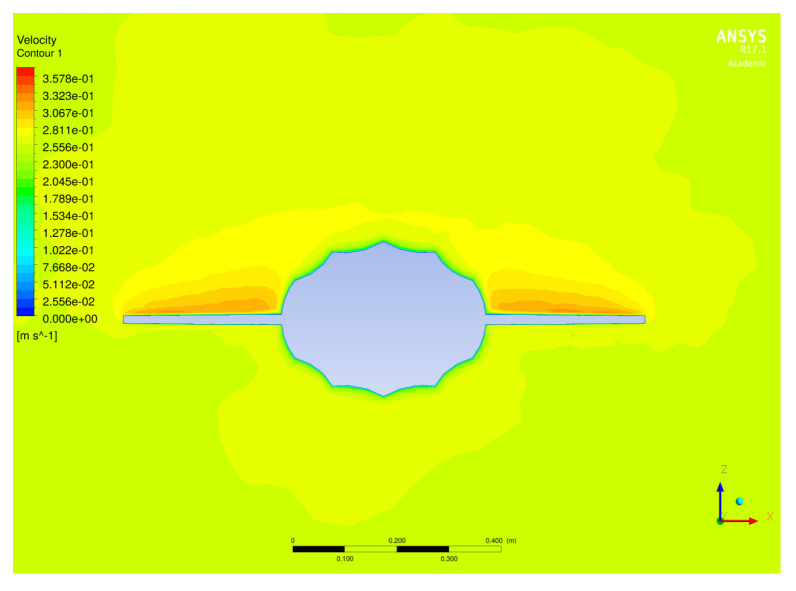
Velocity contour of the glider’s cross-section. Simulations performed using ANSYS Fluent^®^ R17.1 [55].

**Figure 9 biomimetics-08-00080-f009:**
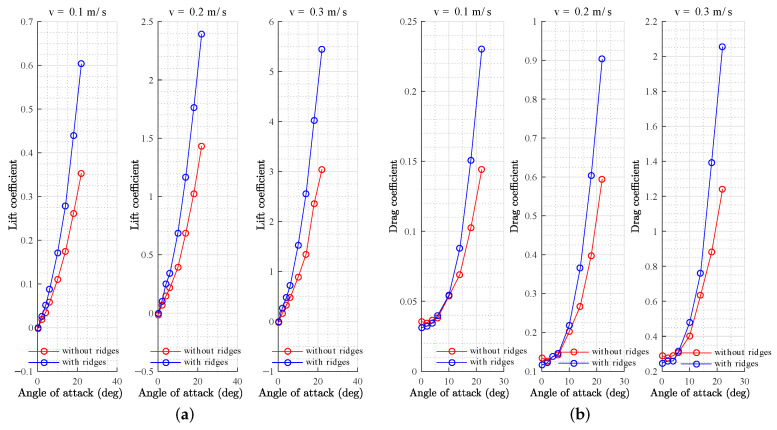
Lift (**a**) and drag (**b**) coefficients at different velocities.

**Figure 10 biomimetics-08-00080-f010:**
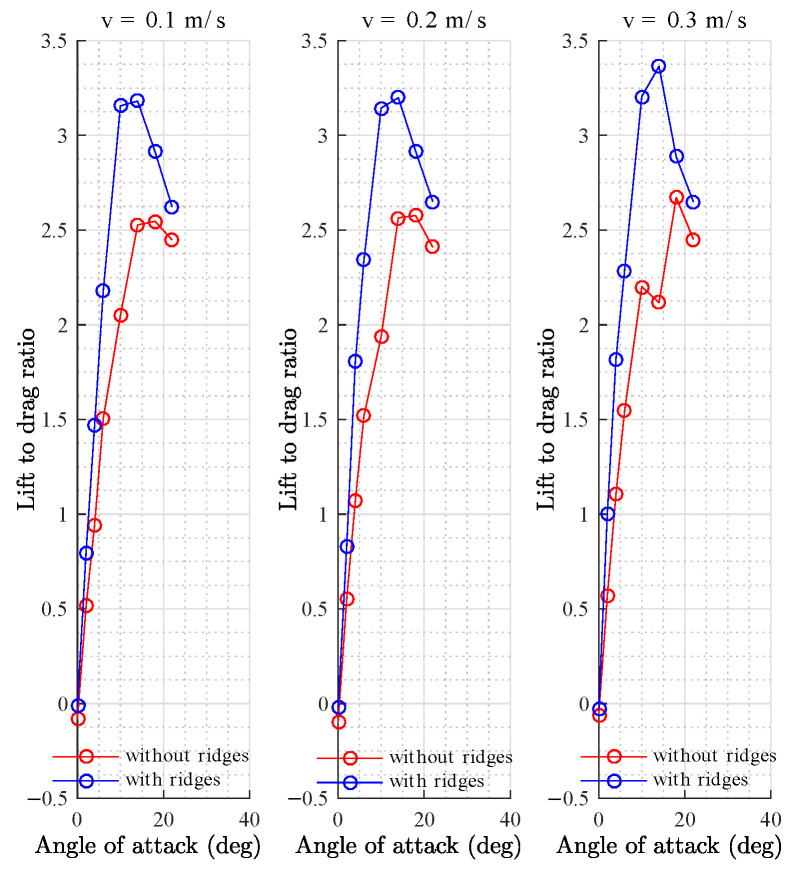
Lift-to-drag ratio at different velocities.

**Figure 11 biomimetics-08-00080-f011:**
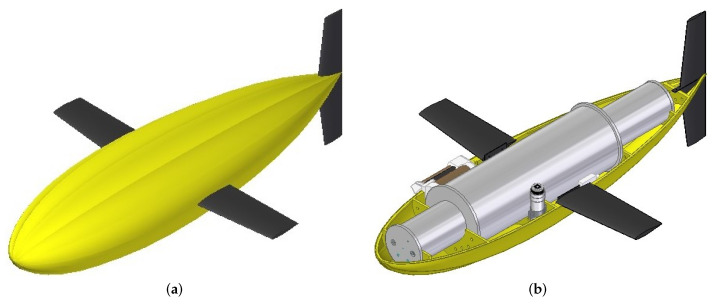
Bioinspired glider layout. (**a**) External layout. (**b**) General layout.

**Figure 12 biomimetics-08-00080-f012:**
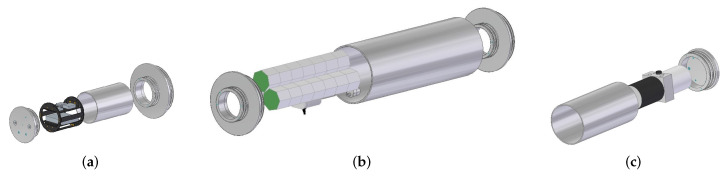
Distribution of electronics (**a**), batteries (**b**), and buoyancy (**c**) bays.

**Figure 13 biomimetics-08-00080-f013:**
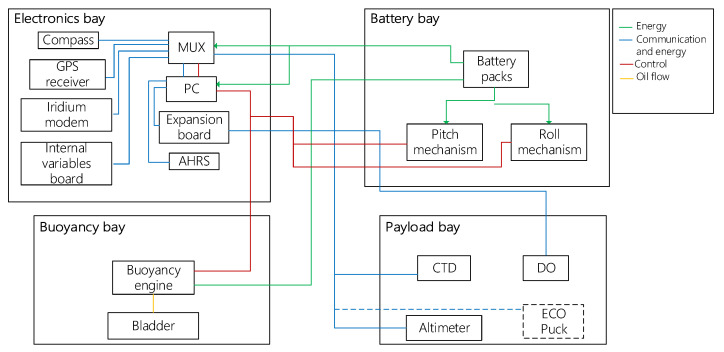
Physical architecture.

**Figure 14 biomimetics-08-00080-f014:**
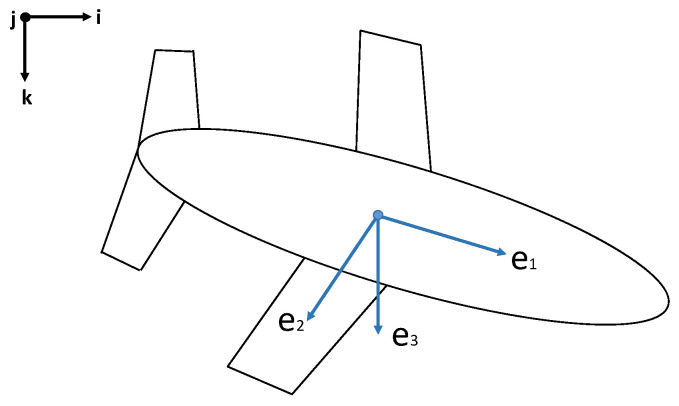
Coordinate system for the underwater glider.

**Figure 15 biomimetics-08-00080-f015:**
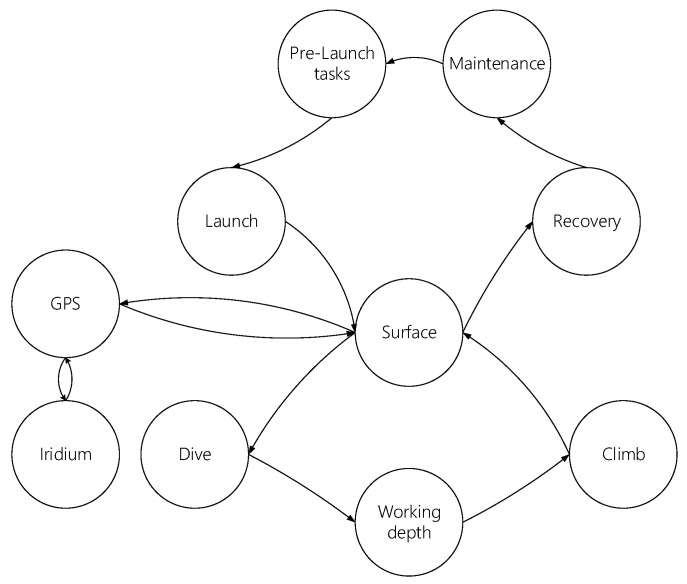
Sequential description of a mission.

**Figure 16 biomimetics-08-00080-f016:**
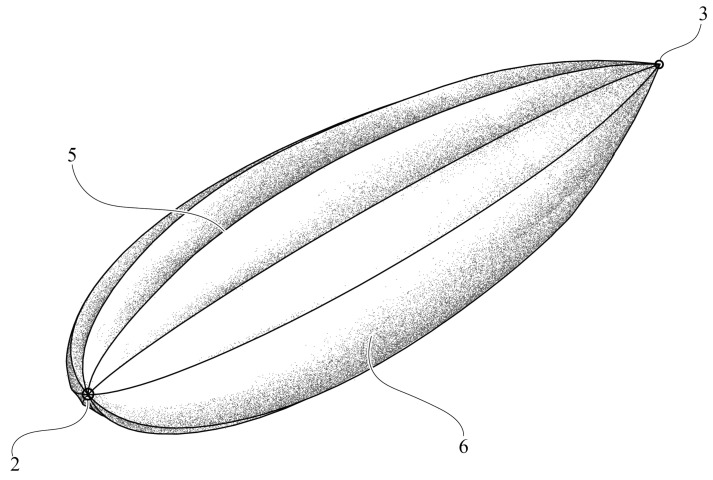
Bioinspired shell. Colombian Patent NC2019/000840 [78].

**Table 1 biomimetics-08-00080-t001:** Wall thickness given by yield and buckling criteria for each hull.

	Hull 1	Hull 2	Hull 3
	(mm)	(mm)	(mm)
Diameter	140	225	140
Length	215	665	300
$t_{y}$	1.015	1.63	1.015
$t_{bc}$	2.295	4.793	2.623
*t*	2.65	4.8	2.65

**Table 2 biomimetics-08-00080-t002:** Electronics bay’s components.

Component	Model	Manufacturer	Voltage	Power Consump.	Comm. Interf.
AHRS	UM7	CH Robotics	5 VDC	0.25 W	SPI
Compass	OS5000-S	Ocean Server	3.3–5 VDC	0.099 W	RS-232
GPS receiver	GPS 15xL	Garmin	3.3–5.4 VDC	0.5 W	RS-232
Iridium transceiver	9603 SBD	Iridium	5 VDC	1 W	RS-232
Embedded computer	Fox	VersaLogic	5 VDC	5.5 W	
Electronic board	N/A	In-house [60]	5 VDC	0.5 W	

## Data Availability

Not applicable.

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
