# Peer review of "Design of a Bioinspired Underwater Glider for Oceanographic Research"

_biomimetics, 2023, doi:10.3390/biomimetics8010080_

Round 1

Reviewer 1 Report

The paper carefully details the design of a bio-inspired underwater glider. The vehicle has been sized according to data available from the literature and numerical predictions obtained through computational fluid dynamics analysis. The authors expoited biomimetics to design the external shape of the glider in order to optimize its hydrodynamic performance and improve its endurance. The potential benefits of the bio-inspired architecture are soundly provided and the design steps are carefully explained. A virtual prototype is the output of the present work and I look forward to seeing the manufactured glider in a future work.

Here are my suggestions to improve the quality of the paper even more. 

In order to underline the consistency of the article with the topics covered by the journal, I suggest to cite in the bibliography other works published in MDPI Biomimetics in areas consistent with the topics covered by the article under review. In particular, are proposed the following papers: 

Design of a Carangiform Swimming Robot through a Multiphysics Simulation Environment by Daniele Costa,Giacomo Palmieri,Matteo-Claudio Palpacelli,David Scaradozzi andMassimo Callegari, Biomimetics 2020, 5(4), 46; 30 Sep 2020

Bioinspired Propulsion System for a Thunniform Robotic Fish by Iliya Mitin,Roman Korotaev,Artem Ermolaev,Vasily Mironov,Sergey A. Lobov andVictor B. Kazantsev, Biomimetics 2022, 7(4), 215; 28 Nov 2022

-

Line 96: "multidisciplinaryv" -> "multidisciplinary"

Line 100: remove "in"

Line 105: "an" -> "a"

Line 371: remove "and"

-

Figure 3: The caption should explain the meaning of the arrows connecting the blocks

-

Line 234: The design began with a platform area of S = 0.15 m2 as a first approximation.

Why? Where does this number comes from? At least a reference should be provided.

-

Line 250: A taper ratio of 0.7 was assumed

The same. Where does this number comes from? The literature?

-

Figure 5: Are this data the result of a direct measurement from the authors or have they been derived from the literature? I believe that a reference should be added.

-

Figure 6: A reference frame or a couple of axes should be added for the sake of clarity

-

Section 4.2.5: what is the overall lenght l of the vehicle?

-

Section 4.2.6: regarding the CFD analyis. Is the adoption of a turbolence model consistent with the low Reynolds number hypotesis stated in the previous section?

-

Section 4.3 The attitude system is carefully datailed. However, further informations should be provided to show if the restoring moment due to moving ballast device is sufficient to produce the pitch variation necessary to the diving maneuver. For example, the authors could compute the shift between the bouyancy force application point and the vehicle mass center as a function of the linear displacement of the battery system. 

Author Response

We would like to thank the reviewer for the insightful comments provided in order to improve the paper.

Reviewer 2 Report

1. Given the leatherback turtle results, such as those from Ref. [51], the improvements in increasing lift and reducing drag by adding longitudinal ridges over a teardrop hull appears obvious. What will be useful and interesting for readers, either in or outside of the field of biomimetics, are the questions about how the longitudinal ridges be designed and how different designs are compared. For examples, how many ridges should be used? How high should a ridge be? Could a shell of a different cross sectional shape, such as rectangular, with longitudinal ridges, be even better? It seems that the manuscript shows results from only one design and the reason of making and selecting that design was not clearly explained. I think the authors should at least describe some more details in the comparison of ridge designs of different parameters.

2. A good design should consider mission purpose and manufacturing cost. In terms of mission purpose, for example, for the type of operations like a Seaglider, I think the authors should explain what main benefits a glider of slightly improved lift and reduced drag can bring. Is it for completing the mission sooner? How much saving of time can be obtained and why it is significant, since a Seaglider-type operation is expected to be slow? In terms of manufacturing, specific ridge design will involve a more complex mold fabrication process. The actual ridge shapes also need to be adjusted to suit molding requirements, such as no sharp edges. How will these factors affect the cost and the efficiency of the new shell? And how can these issues be dealt with?

Author Response

(The authors gave the same response as above.)

Round 2

Reviewer 2 Report

The authors have generally modified the manuscript in answering the questions from my comments.